



# Strong temporal variation in treefall and branchfall rates in a tropical
# forest is explained by rainfall: results from five years of monthly drone
# data for a 50-ha plot
Raquel Fernandes Araujo[1], Samuel Grubinger[2], Carlos Henrique Souza Celes[1], Robinson I. Negrón-
Juárez[3], Milton Garcia[1], Jonathan P. Dandois[4], and Helene C. Muller-Landau[1]
[1]Center for Tropical Forest Science-Forest Global Earth Observatory, Smithsonian Tropical Research Institute, Balboa, Ancon,
PO Box 0843-03092, Panama
[2]Department of Forest Resources Management, University of British Columbia, 2424 Main Mall, Vancouver, BC V6T 1Z4,
Canada
[3]Climate Sciences Department, Lawrence Berkeley National Laboratory, 1 Cyclotron Road, Berkeley, CA 94720, USA
[4]Johns Hopkins University, Facilities and Real Estate, 3910 Keswick Rd. Suite N3100 Baltimore, MD 21211, USA
*Correspondence to*: Raquel Fernandes Araujo (araujo.raquelf@gmail.com)
**Abstract.** A mechanistic understanding of how tropical tree mortality responds to climate variation is urgently needed to predict
how tropical forest carbon pools will respond to anthropogenic global change, which is altering the frequency and intensity of
storms, droughts, and other climate extremes in tropical forests. We used five years of approximately monthly drone-acquired
RGB imagery for 50 ha of mature tropical forest on Barro Colorado Island, Panama, to quantify spatial structure, temporal
variation, and climate correlates of canopy disturbances, i.e., sudden and major drops in canopy height due to treefalls, branchfalls,
or collapse of standing dead trees. Treefalls accounted for 77 % of the total area and 60 % of the total number of canopy
disturbances in treefalls and branchfalls combined. The size distribution of canopy disturbances was close to a power function for
sizes above 25 m$^2$, and best fit by a Weibull function overall. Canopy disturbance rates varied strongly over time and were higher
in the wet season, even though windspeeds were lower in the wet season.  The strongest correlate of temporal variation in canopy
disturbance rates was the frequency of 1-hour rainfall events above the 99.4$^{th}$ percentile (here 35.7 mm hour$^{-1}$, r = 0.67). We
hypothesize that extreme high rainfall is associated with both saturated soils, increasing risk of uprooting, and with gusts having
high horizontal and vertical windspeeds that increase stresses on tree crowns. These results demonstrate the utility of repeat drone-
acquired data for quantifying forest canopy disturbance rates over large spatial scales at fine temporal and spatial resolution, thereby
enabling strong tests of linkages to drivers. Future studies should include high frequency measurements of vertical and horizontal
windspeeds and soil moisture to better capture proximate drivers, and incorporate additional image analyses to quantify standing
dead trees in addition to treefalls.

**1 Introduction**
Tropical forests account for two-thirds of terrestrial biomass carbon stocks (Pan et al., 2013), and uncertainty regarding
the future of these stocks is a major contributor to uncertainty in the future global carbon cycle (Cavaleri et al., 2015). Tropical



forest carbon stocks depend critically on tree mortality rates, and theory and evidence suggest tropical tree mortality rates may be
increasing due to anthropogenic global change (Brienen et al., 2015; McDowell et al., 2018). Tropical tree mortality can be caused
by a diversity of drivers including storms, droughts, fires, lightning strikes, and biotic agents (McDowell et al., 2018; Yanoviak et
al., 2017; Fontes et al., 2018; Silva et al., 2018). The frequency of extreme rainfall and drought events is expected to increase in
tropical regions, potentially increasing associated tree mortality (IPCC, 2014; Deb et al., 2018; Aubry-Kientz et al., 2019). An
improved understanding of the processes of forest disturbance is critical to constrain estimates of current and future carbon cycling
in tropical forests under alternative emissions scenarios (Leitold et al., 2018).
Despite the importance of tree mortality to forest structure and carbon turnover rates, the mechanisms underlying tree
mortality remain unclear (McDowell et al., 2018). A key problem is that remeasurement intervals of permanent plots average five
or more years, making it difficult to link mortality variation with particular climatic events (Phillips et al., 2010; Davies et al.,
2021; Arellano et al., 2019).  The high rates of decomposition in tropical forests further obscure evidence of underlying mechanisms
and risk factors (Arellano et al., 2019). The few studies that have quantified temporal variation of tree mortality at monthly and bi-
monthly scales using ground-based data have all found higher tree mortality in times of higher rainfall (Brokaw, 1982; Fontes et
al., 2018; Aleixo et al., 2019). This is consistent with the understanding that many trees die in treefalls, which are proximately
caused by trunk breakage or uprooting, and are associated with storms (Marra et al., 2014; Araujo et al., 2017; Fontes et al., 2018;
Negrón-Juárez et al., 2018; Esquivel-Muelbert et al., 2020). The collection of additional high temporal resolution mortality data
over large areas, together with high temporal resolution climatological data, can aid in linking mortality to particular climatic
events and thereby elucidating mortality mechanisms (Arellano et al., 2019; McMahon et al., 2019).
Drone-acquired aerial imagery and photogrammetry software now provide excellent tools for monitoring forest canopies
(Araujo et al., 2020) and repeat drone flights can quantify canopy dynamics over large areas at high temporal resolution.
Photogrammetric analysis of simple RGB imagery enables reconstruction of the appearance and three-dimensional structure of the
top of the canopy at high spatial resolution (Dandois and Ellis, 2013; Araujo et al., 2020; Zahawi et al., 2015). Comparison of
photogrammetry products from successive drone flights allows easy detection and quantification of treefalls and branchfalls of
canopy trees. Canopy trees constitute a high proportion of stem density, aboveground carbon stocks and wood productivity (Araujo
et al., 2020), and thus information on their dynamics is disproportionately useful.  Treefalls do not necessarily result in tree
mortality (trees may survive and resprout), but all treefalls and branchfalls result in a large flux of carbon (wood) from biomass to
necromass, i.e., biomass turnover, which translates to reduced woody residence time. Periods of higher canopy disturbance rates
thus represent periods of higher biomass turnover, and likely correlate with higher tree mortality rates.  Further, even when trees
don't die from a canopy disturbance event, suffering crown loss or damage increases the risk of subsequent mortality (Arellano et
al., 2019).
Monitoring canopy disturbances with drones also provides the opportunity to precisely quantify the size distributions of
these canopy disturbances, and to distinguish branchfalls from treefalls. Here we define a canopy disturbance as a substantial
decrease in canopy height in a contiguous patch of canopy occurring over one measurement interval, such as typically results from
a treefall or branchfall.  Marvin and Asner (2016) and Dalagnol et al. (2021) referred to these as "dynamic canopy gaps." By
definition, canopy disturbances reduce canopy height and thereby change light regimes for understory and neighboring trees, and
the magnitude of the change depends on the disturbance size in area and depth (Hubbell et al., 1999). In general, larger canopy
disturbances cause larger canopy gaps as traditionally measured on the ground. Previous studies have analyzed the size distributions
of static gaps for insights into forest structure, habitat niches, and disturbance regimes (e.g., Manrubia and Solé, 1997; Lobo and
Dalling, 2013, 2014; Fisher et al., 2008). Tree species respond differently to canopy gaps of different sizes, with small gaps favoring





a different set of species than large gaps (Brokaw, 1985; Denslow, 1980, 1987; Dalling et al., 2004). Branchfalls, like treefalls, are
important in generating canopy gaps and contributing to woody turnover, but also often go unmeasured (Marvin and Asner, 2016;
Leitold et al., 2018). Quantifying tree mortality and other non-fatal damage such as branchfall thus contributes to a better
understanding on change of forest structure, necromass estimates and nutrient cycling.
Here, we use 5 years of ~monthly drone-acquired RGB imagery for a 50 ha area of mature tropical forest on Barro
Colorado Island, Panama, to investigate canopy dynamics at high temporal resolution. We aim to (1) quantify temporal variation
in canopy disturbance rates and its relationship to climate variation; (2) characterize the size structure of canopy disturbances; and
(3) evaluate the role of branchfalls in canopy dynamics. We expect that disturbance rates will be higher in the wet season than the
dry season, and will increase with the frequency of extreme rainfall and wind events, and we compare models differing in the
conditions for defining such extreme events. To characterize the size structure of canopy disturbances, we quantify the size (area)
distribution and evaluate whether it is best fit by power, Weibull, or exponential functions. Finally, we quantify the proportion of
canopy disturbance due to branchfalls (rather than treefalls), and test whether branchfalls and treefalls exhibit similar patterns of
temporal variation. Our results provide new insights into the patterns and drivers of canopy disturbance and tree mortality in this
tropical forest, and illustrate the utility of drones for quantifying canopy dynamics over large areas at high temporal resolution.

**2. Methods**

**2.1 Study site**
Barro Colorado Island (BCI; 9°9′ N, 79°50′ W) is a 15 km$^2$ island in Central Panama, that was isolated from surrounding
mainland when Lake Gatun was created as part of the construction of the Panama Canal. BCI supports tropical moist forest in the
Holdridge Life Zone System (Holdridge, 1947). Annual precipitation averages approximately 2600 mm, with a pronounced dry
season between January and April (a mean of about 3.5 months with < 100 mm mo$^{-1}$). Mean annual temperature is 26 °C and varies
little throughout the year (Windsor, 1990). The 50 ha forest dynamics plot (1000 m x 500 m) was established on BCI in 1981
(Hubbell et al., 1999). It is located in old-growth forest (Leigh, 1999), with the exception of a small area of 1.92 ha of old secondary
forest (~100 years old) in the north central part of the plot (Harms et al., 2001).

**2.2 Meteorological data**
Meteorological data were collected in the lab clearing and Lutz tower, approximately 1.7 km NE of the center of the 50
ha plot. Wind speed was measured using an anemometer (RM Young Wind Monitor Model 05103) installed at the top of Lutz
tower, at 48 m height above ground and approximately 6 m above the top of the surrounding canopy. The maximum wind speed
was recorded for every 15 minute-interval. Rainfall was measured in the lab clearing using a tipping bucket (Hydrological Services
Model TB3), and recorded every 5 minutes; we aggregated these data to 15-minute periods to match the temporal resolution of the
wind speed data. Rainfall and wind speed data are available in
https://biogeodb.stri.si.edu/physical_monitoring/research/barrocolorado. The meteorological record had no gaps during our study
period (Fig. S1).




### 2.3 Canopy disturbance identification

We used approximately monthly orthomosaics and canopy surface models produced from drone-acquired imagery to analyze temporal variation in canopy disturbance rates in the 50 ha plot between 2 October 2014 and 28 November 2019. RGB imagery was collected using a variety of drones and cameras over the years, with a horizontal spatial resolution of 3-7 cm. Imagery for each sampling date was processed using the photogrammetry software Agisoft Metashape to obtain orthomosaics and surface elevation models, which were then aligned vertically and horizontally (details in Text S1).

We defined a canopy disturbance as a substantial decrease in canopy height in a contiguous patch of canopy occurring over one measurement interval, such as typically results from a treefall or branchfall. We identified canopy disturbances through a combination of analysis of the canopy surface model changes and visual interpretation of the orthomosaics (Fig. 1). We first differenced surface elevation models for successive dates to obtain a raster of the canopy height changes for the associated interval (Fig. 1, Text S1). We then pre-delineated major canopy disturbances by filtering for areas in which canopy height decreased more than 10 m in contiguous areas of at least 25 m$^2$ (the minimum area for canopy gaps in previous studies by Brokaw (1982) and Hubbell et al. (1999)), and that had an area-to-perimeter ratio greater than 0.6. (The area-to-perimeter condition removes artifacts associated with slight shifts in the measured positions of individual trees from one image set to another, whether due to wind or alignment errors.) Finally, we systematically examined orthomosaic images for 1-ha square subplots for each pair of successive dates and edited the pre-delineated polygons, removed false positives, and added visible new canopy disturbances that were not previously delineated (whether because they were too small in area or in canopy height drop). During the visual inspection of the data for the last three years we also classified disturbances as being due to treefalls (a whole previously live tree fell, creating a clearly visible gap on the forest floor, or the whole live crown disappeared), branchfalls (a portion of a live crown broke), or standing dead trees disintegrating (Fig. S2).

132



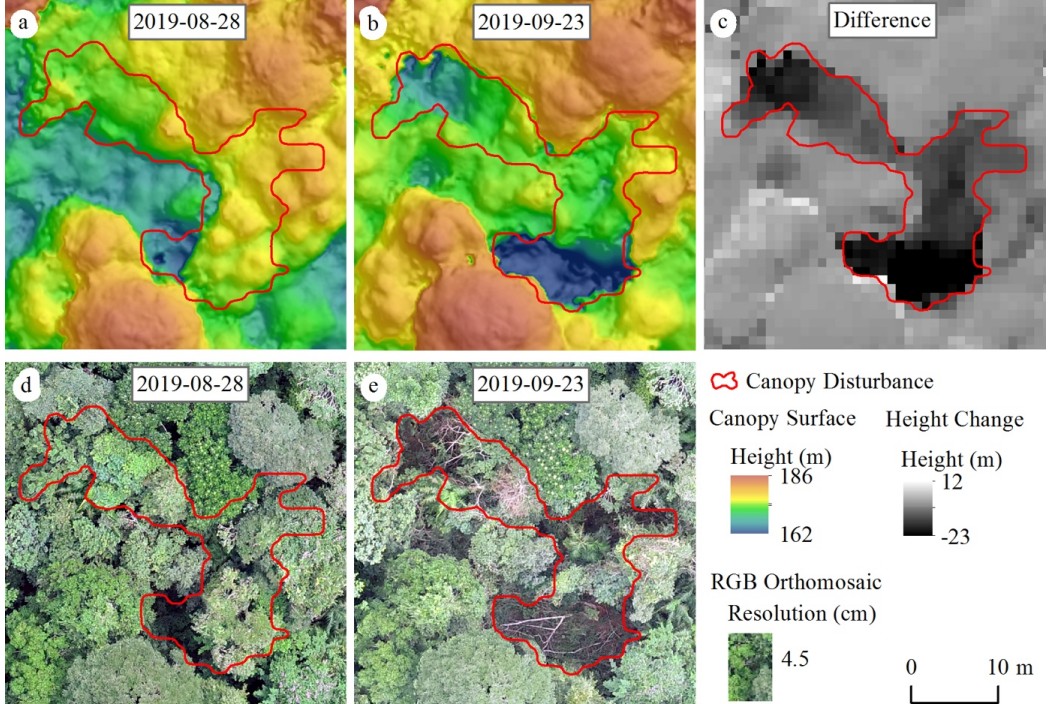

**Figure 1.** Canopy disturbance visualized on canopy surface models and orthomosaics calculated from photogrammetric analyses of drone imagery. (a,b) Elevation models for a portion of the study area on two successive dates, 28 August 2019 (a) and 23 September 2019 (b). (c) Difference in elevation between the two dates, with black area indicating large decrease in canopy elevation. (d,e) RGB orthomosaics of the same dates.

We calculated the total number and area of canopy disturbances within the BCI 50 ha plot during the five years of the study. In calculating the number and total area of disturbances, we included all disturbed areas that were inside the plot boundaries (if a disturbance was on the boundary, only the area inside the plot was included). Our analyses of temporal variation employed the same definitions for numbers and areas of canopy disturbances within the 50 ha plot. For analyses of the size structure of disturbances, we included the complete areas of disturbances whose centroids were located within the plot (i.e., we excluded disturbances centered outside the plot, and included area outside the plot for disturbances centered inside the plot to avoid artifacts related to reducing disturbance size by trimming at the plot boundaries).

**2.4 Temporal variation in canopy disturbance rates and its relation to climate**

We calculated canopy disturbances rates for each measurement interval as the percentage of area disturbed per month (i.e., per 30-day period). Specifically, we summed the total area disturbed during the measurement interval, and divided by the total area of the plot and the length of the time interval. We excluded one excessively long interval (237 days) from all analyses of temporal variation; the remaining intervals ranged from 14 to 91 days, with a median of 31.5 days (Table S1). We also calculated an incidence canopy disturbance rate as the *number* of canopy disturbances per hectare per month. We calculated the mean,





minimum, maximum, and the 25th, 50th, and 75th percentiles of interval length in days, number and area of canopy disturbances,
and the respective monthly rates.
We compared canopy disturbance rates between wet and dry seasons and between early wet and late wet seasons. We
defined the dry season as January 1 to April 30 (rainfall < 100 mm mo$^{-1}$, Fig. S3), the early wet season as 1 May to 31 August, and
the late wet season as 1 September to 31 December. Intervals that straddled more than one season were classified to the season in
which they had more days. We tested for homogeneity of variances using the Levene test, and for differences between means using
the two-tailed Student's t-test for the log-transformed canopy disturbance rates.
We evaluated the relationship of temporal variation in canopy disturbance rates with temporal variation in climate
extremes using linear regressions. We regressed canopy disturbance rates (area per time) against the frequency of extreme rainfall
and windspeed events (number per time), for different definitions of extreme events. For example, one definition of an extreme
event would be a 15-minute period with rainfall above the 99th percentile.  We evaluated three different temporal grains for defining
extreme events (15-minute, 1-hour, and 1-day intervals), for two different meteorological variables (total rainfall and maximum
windspeed), and 100 different thresholds, corresponding to every 0.1 percentile increment between the 90th and 99.9th percentile
of the corresponding distributions. We compared the predictive ability of these 600 different definitions of extreme events in terms
of their Pearson correlations.

### 2.5 Size structure of canopy disturbances

We characterized the size structure of canopy disturbances whose geometric center was inside the plot, excluding
disturbances from the one excessively long interval of 237 days. (Longer time intervals increase the likelihood that what is
measured as a single disturbance event in fact constitutes multiple adjoining or overlapping events.) We calculated the mean,
minimum, maximum, and median of area of individual canopy disturbances.  We graphed the cumulative distribution functions
with respect to individual disturbance area of number and area of canopy disturbances, to quantify the proportions of canopy
disturbances and of total area disturbed below any given size.
We took advantage of the three-dimensional structure of our photogrammetry data to quantify canopy disturbances in
terms of their vertical height drop as well as their horizontal area.  For each canopy disturbance, we calculated the average height
drop from the differences in the canopy surface models. (We excluded 61 canopy disturbances in which heights increased because
they reflect errors in the canopy height models.) We evaluated how average height drop was related to area across canopy
disturbances, graphically and in terms of their Pearson correlation.
We quantified the size distributions of canopy disturbances by fitting three alternative probability distributions:
exponential, power, and Weibull.  Recognizing that our methods may miss smaller disturbances, we fit these distributions to
truncated datasets, excluding disturbances below 2, 5, 10 or 25 m$^2$. (Note that 25 m$^2$ is the minimum area for defining a canopy
disturbance in our automated pre-delineation algorithm, and we are confident we captured all disturbances above this area.)  We
binned the data into 1 m$^2$ classes, and fitted each distribution to each truncated dataset using maximum likelihood, as described in
(Araujo et al., 2020). We compared the goodness of fit of the different functions using Akaike's Information Criterion (AIC).






### 2.6 Branchfalls vs. treefalls

188       For the last three years, for which we classified each canopy disturbance as being a branchfall, treefall, or standing dead
tree, we evaluated the relative contributions of branchfalls vs. treefalls. We did not include standing dead trees in the analysis
because our methods possibly missed many standing dead trees. We separately calculated treefall and branchfall disturbance rates
for each interval, and relative contributions to their summed number and area. We regressed branchfall disturbance rates against
treefall disturbance rates, for both area- and number-based rates, and calculated their Pearson correlations.

### 3. Results

195       We identified 1056 canopy disturbances with a combined area of 56,595.12 m$^2$ that affected the area within the BCI 50
ha plot between 2 October 2014 and 28 November 2019 (Fig. 2). During the 5 years of the study, 10.7 % of the area of the BCI
50-ha plot was affected by canopy disturbances, and 0.7 % was disturbed more than once (Fig. 2).

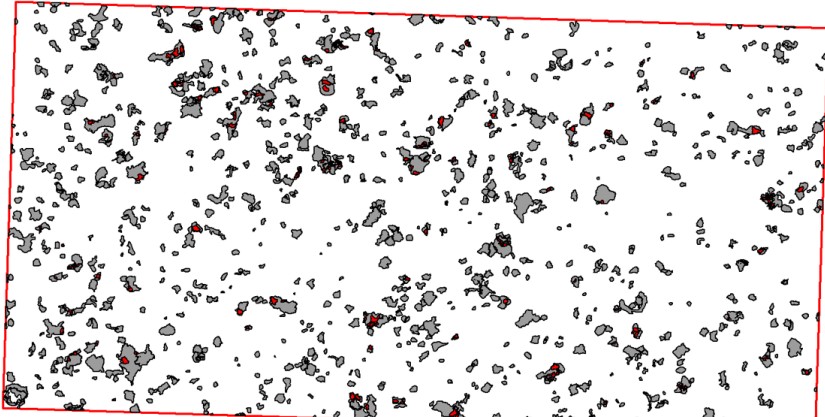


**Figure 2.** Map of canopy disturbances on the 50 ha plot (red rectangle, 1000 x 500 m) on Barro Colorado Island, Panama, from 2
October 2014 to 28 November 2019. Areas that were disturbed a single time are shown in grey, those disturbed more than once
in red.


### 3.1 Temporal variation in canopy disturbance rates

204       Temporal variation analyses included 906 disturbances or partial disturbances encompassing 50,202.8 m$^2$ of area that
were located inside the 50 ha plot and were not part of the excluded long interval. There was strong temporal variation in canopy
disturbance rates among the 46 time intervals analyzed, with parallel variation in the total area disturbed (Fig. 3) and the number
of disturbances (Fig. S4). The mean rate of canopy disturbance creation was 916 m$^2$ mo$^{-1}$ (range of 75 m$^2$ mo$^{-1}$ to 8040.9 m$^2$ mo$^{-1}$)



and median 499 m² mo⁻¹ (other statistics in Table S1).
The highest disturbance rates occurred during May-July 2016, May-August 2018, and August-September 2019 (Fig. S5).
The single highest disturbance rate was observed between 1 June and 13 July 2016, when 11,257 m² of disturbances were created
in just 42 days (a rate of 268 m² day⁻¹). A full 2.3 % of the total area of the plot was converted to new canopy disturbances during
this time interval. In contrast, the total area of new disturbances across the rest of the 5-year period was 38,946 m² (a rate of 24.3
m² day⁻¹).

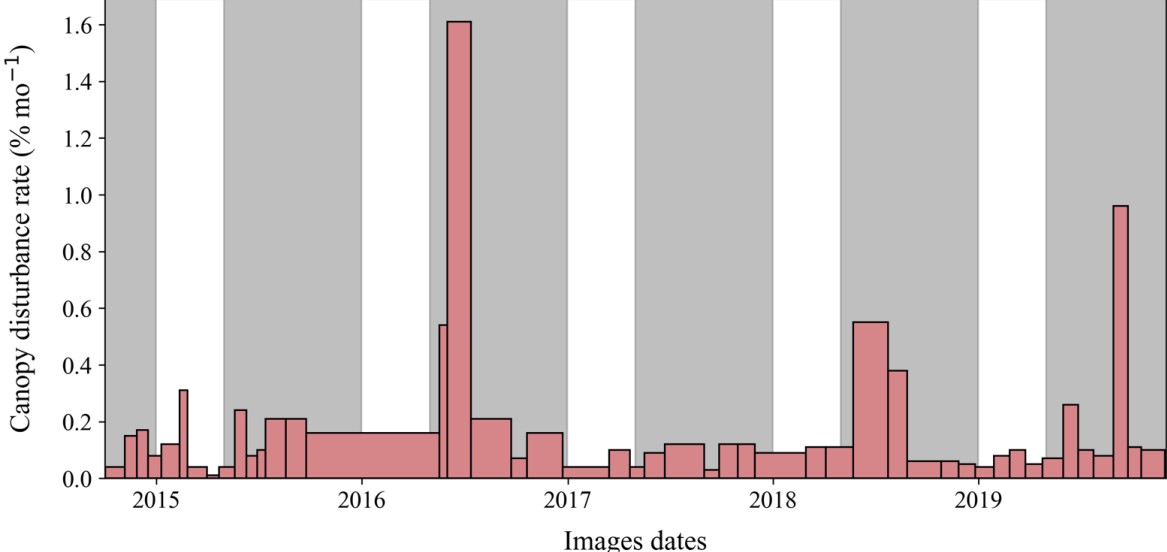


**Figure 3.** Temporal variation in canopy disturbance rates in the 50 ha plot on Barro Colorado Island, Panama, across measurement
intervals. Gray shading indicates the wet seasons (May to December) of each year and ticks on the x axis indicate the first day of
each year. Rates are shown in units of percent of area per month (30-day period). Note that the total area of each rectangle is
proportional to the total area of canopy disturbed during that measurement interval.
Rates of canopy disturbances were higher during the wet season (p = 0.03; Fig. 4a). There was no significant difference
in rates between the early and late wet season (p = 0.27, Fig. 4b). Very high rates of disturbance (> 0.3 % per month) were observed
only in the wet season.





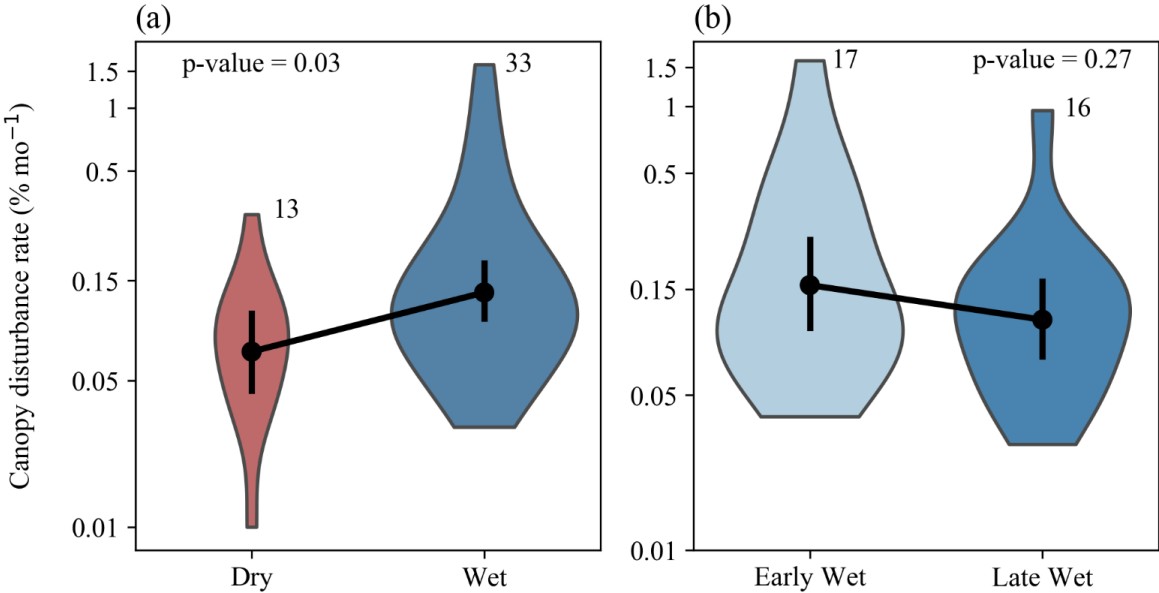


**Figure 4.** Comparisons of canopy disturbances rates between wet and dry seasons (a), and between early and late wet seasons (b). Violin plots depict the distributions of disturbance rates (% area disturbed per month) over time intervals, with the number of time intervals listed above each violin plot. Black dots and bars show means and 95% confidence intervals, respectively.

The best predictor of temporal variation in canopy disturbance rates was the frequency of 1-hour rainfall events above the 99.4th percentile, here 35.7 mm hour$^{-1}$, which explained 45 % of the variation (Fig. 5a). This threshold outperformed all other tested rainfall thresholds (all percentiles from 90.0 to 99.9, by 0.1 % of the different frequency time scales – Fig. 5b). Only two of these high rainfall events occurred during the same day (Table S2). The measurement interval with the highest disturbance rate (June 1 to July 13 2016) included four such high rainfall events: 41.7 mm hour$^{-1}$ on June 17, 41.9 mm hour$^{-1}$ on June 23, 49.3 mm hour$^{-1}$ on June 30, and 36.1 mm hour$^{-1}$ on July 4 (Table S2). The frequency of high horizontal maximum wind speed events was not significantly related with canopy disturbance rates. Indeed, Pearson correlations were negative for almost all wind speed variables (Fig. S6).



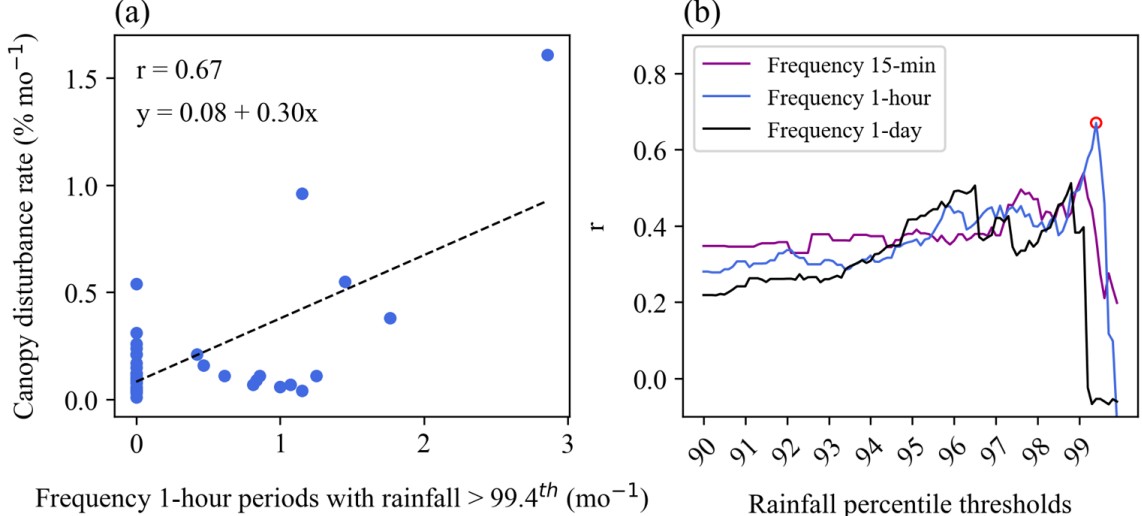

234

**Figure 5.** Relation of temporal variation in canopy disturbance rates to the frequency of extreme rainfall events. (a) The relationship
for the single best predictor of canopy disturbance rate: the frequency of 1-hour periods with rainfall exceeding the 99.4th percentile;
each point represents one measurement interval, and the dashed line shows the linear regression. (b) Variation in Pearson
correlation between canopy disturbance rate and frequency of extreme rainfall events depending on the temporal grain (colors) and
percentile threshold (x axis) for defining extreme rainfall events; the open red circle indicates the best correlation.


## 3.2 Size structure of canopy disturbances

A total of 878 canopy disturbances with 49,958 m$^2$ total area had their centers inside the plot and were not part of the
excluded long interval, and thus were included in the size distribution analyses. The areas of mapped individual canopy
disturbances ranged from 2.2 m$^2$ to 486.7 m$^2$, with a mean of 56.9 m$^2$. The median disturbance area was 36.4 m$^2$, whereas 50 % of
the total area was in disturbances greater than 86.6 m$^2$ (see Fig. 6a for the full cumulative distributions by gap number and area).
Canopy disturbances with larger areas tended to have larger mean decreases in canopy height (Pearson r = 0.39, Fig. 6b).
The size distribution of canopy disturbances was close to a power function for areas above 25 m$^2$, and was relatively flat
over the range of 5 to 25 m$^2$ (Fig. 6c). The fitted exponent of the power function was -1.96 for canopy disturbances above 25 m$^2$,
but the Weibull distribution provided a better fit than the power function (Table 1). When distributions were fit to data including
smaller size classes (> 2 m$^2$, > 5 m$^2$ or > 10 m$^2$), the distribution is further from a power function; the Weibull remains the best fit,
the exponential becomes the second-best fit, and the power function the worst fit of the three (Fig. S7, Table S3).





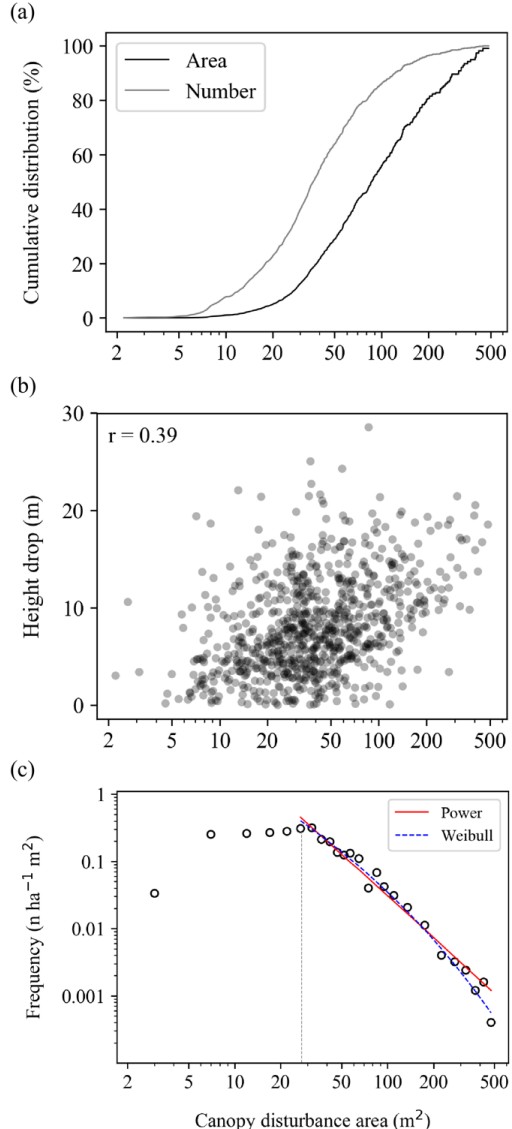


**Figure 6.** Size structure of canopy disturbances. (a) Cumulative number and area of canopy disturbances in relation to their area. (b) Relationship of mean vertical height drop to horizontal area among canopy disturbances. (c) Size distribution of canopy disturbances, together with Weibull and power function fits for canopy disturbances larger than 25 m$^2$ (this threshold was chosen because we are confident we identified all canopy disturbances above this area, but we may have missed some smaller ones). The vertical dashed gray line indicates the 25 m$^2$ threshold.







**Table 1.** Parameter values and delta AIC values for maximum likelihood fits of exponential, power and Weibull probability density
functions to size distributions for canopy disturbances larger than 25 m$^2$. Delta AIC is the difference in AIC from the best model.
The best-fit model is highlighted in bold.

| Distribution | $\lambda$ | $k$ | Delta AIC |
|---|---|---|---|
| Exponential | 0.020 | | 62.45 |
| Power | 1.963 | | 16.50 |
| **Weibull** | **6.745** | **0.448** | **0.00** |



### 3.3 Treefalls and branchfalls

A total of 411 canopy disturbances with 23,289.9 m$^2$ total area occurred during the final three years, and thus were
included in the analyses of branchfall contributions. Branchfalls accounted for 23 % of the total area and 40 % of total number of
disturbances in treefalls and branchfalls combined. Treefall and branchfall disturbance rates varied largely in parallel (Fig. 7, Fig.
S8). Branchfalls were a larger proportion of events and area in some measurement periods than others. The ratio of area in
branchfalls to area in treefalls ranged from 0.07 to 1.4 among measurement periods (Fig. 7a), and the ratio of number of branchfalls
to number of treefalls ranged from 0.2 to 2.3 (Fig. 7b). Standing dead trees accounted for 6.6 % of the total number and 6.7 % of
the total area of mapped canopy disturbances.

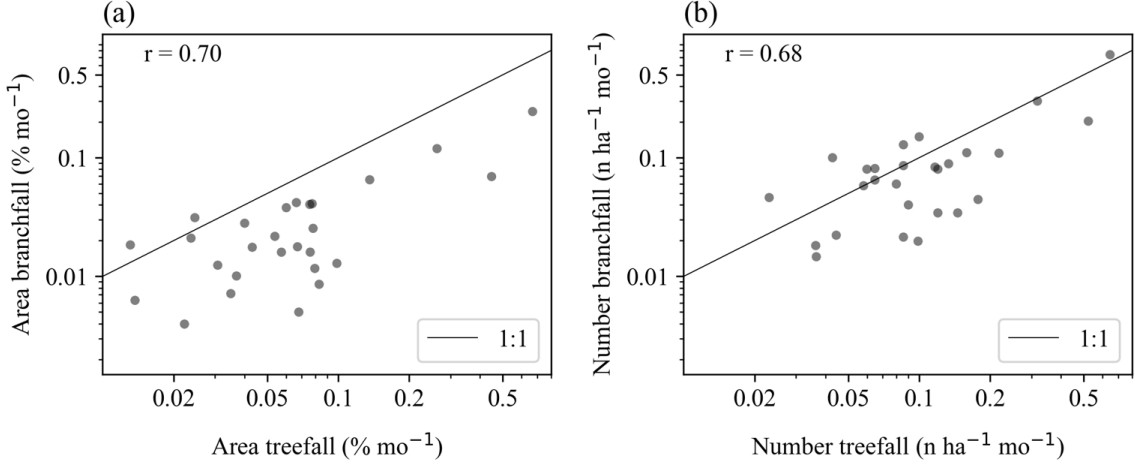


**Figure 7.** Relationship of temporal variation in branchfall rates to temporal variation in treefall rates, when measured by total area
(a) and number of events (b). This includes measurement intervals from 23 December 2016 to 28 November 2019.



## 4. Discussion

The use of high frequency (approximately monthly) drone imagery enabled us to quantify temporal variation in canopy disturbance rates and to quantify the sizes of canopy disturbances at high temporal and spatial resolutions. We found that canopy disturbance rates of the BCI 50 ha plot varied strongly over time, and were higher in the wet season. The frequency of extreme rainfall events was the single best predictor of monthly variation in canopy disturbance rate during the 5-year study period. In contrast, maximum horizontal wind speed was not significantly related. The size distribution of canopy disturbances was close to a power function for larger canopy disturbances, but best fit by a Weibull function overall. Branchfalls accounted for 23 % of the total area of disturbances from treefalls and branchfalls combined, and branchfall rates varied largely in parallel with treefall rates over time. These findings contributed to an improved understanding of the size distribution, temporal variation and meteorological drivers of canopy disturbances in tropical forests.

### 4.1 Temporal variation in canopy disturbance

Canopy disturbance rates varied strongly over time in this moist tropical forest, and were higher in the wet season. A single time interval (June 1 to July 13 2016) accounted for 21 % of the total disturbed area of the BCI 50-ha plot. Treefall and branchfall disturbance rates varied largely in parallel, but not entirely. Some of the differences in temporal patterns simply reflect the stochastic nature of these processes, but different temporal patterns in branchfalls vs. treefalls could also reflect different sensitivity to particular abiotic drivers (e.g. wind regime, soil saturation). The frequency of rainfall events > 35.7 mm hour$^{-1}$ explained much of the variation in canopy disturbance rates among measurement intervals, whereas the frequency of high maximum horizontal wind speeds was not related. At our site, horizontal wind speeds are higher during the dry season, when canopy disturbance rates are lower (Fig. 4a, Fig. S1). We hypothesize that extreme high rainfall is associated with both saturated soils, increasing risk of uprooting, and with gusts having high horizontal and vertical windspeeds that increase stresses on tree crowns. Future studies should include high frequency measurements of vertical and horizontal windspeeds and soil moisture to better capture proximate drivers, and evaluate mechanistically formulated predicted models that include multiple variables.

These results are consistent with previous findings on seasonal variation and the role of rainfall in gap formation in tropical forests. A previous 4-year study on BCI found seasonal peaks in August and September, in the middle of the wet season, with monthly treefall rates significantly correlated with rainfall (r = 0.47, p < 0.02) (Brokaw, 1982). Tree mortality was also strongly and positively correlated with monthly rainfall (r = 0.85) in a 1-year study of a 10-ha site in the Central Amazon (Fontes et al., 2018). A study monitored canopy trees monthly over five decades in the Central Amazon and found that trees died more often during wet months, even in drought years (Aleixo et al., 2019). A regional study of the Central Amazon based on 12 years of satellite data found that major windthrows (visible on LANDSAT) occurred more frequently between September and February, months characterized by heavy rainfall, than the rest of the year (Negrón-Juárez et al., 2017). Analysis of spatial variation in forest damage from Hurricane María in Puerto Rico found that total rainfall was the most important meteorological risk factor and maximum sustained one-minute wind speeds the second-most-important; these two variables were moderately correlated (r = 0.43) (Hall et al., 2020).

Multiple studies have highlighted the importance of mesoscale convective systems, such as squall lines, for windthrows (Garstang et al., 1998; Negrón-Juárez et al., 2010, 2017; Araujo et al., 2017). In Panama, the period of June to August has the higher number of mesoscale convective systems (Jaramillo et al., 2017), and these were the months when we observed the highest



canopy disturbance rates. The threshold rainfall rate of 35.7 mm hour$^{-1}$, which defined the extreme rainfall rate that was the best
predictor of canopy disturbance formation in our study, is six times higher than the mean rate for mesoscale convective systems in
the Panama region (Jaramillo et al., 2017), highlighting the importance of extreme events.

### 4.2 Mechanisms and size structure canopy disturbances

Gaps in the forest canopy can be caused proximally by treefalls of canopy trees, branchfalls of canopy branches, standing
dead canopy trees, or senescing major canopy branches. Treefalls and branchfalls of canopy trees are well-captured in our analyses,
which focus on short-term changes that indicate loss of major canopy elements.  In contrast, standing dead trees and senescing
branches generally involve more subtle changes in the canopy over a longer period of time, and may be missed by our methods.
Treefalls account for a majority of canopy tree mortality in most tropical forests, but standing tree mortality also plays a major
role, especially in drought periods. Overall, treefalls (in which trees were uprooted or their trunks snapped) accounted for 51.2 %
of all mortality of trees > 10 cm DBH in a large-scale study of tree mortality in 189 Amazonian plots  (Esquivel-Muelbert et al.,
2020) and 65 % in a study that monitored tree mortality in 10 ha of forest in the Central Amazon bi-monthly over one year (Fontes
et al., 2018). Treefalls can involve a single canopy tree, or multiple canopy trees. Multi-tree treefalls can result from coordinated
disturbances over a large area (e.g., large footprint wind disturbance) and/or from domino effects in which the failure of one canopy
tree directly stresses one or more neighboring trees and causes them to fall as well (e.g., when additional trees are knocked down
by the first tree, or pulled down because of connections via lianas). It has been hypothesized that canopy disturbances may also be
contagious over longer time intervals, with increased risk of treefall near canopy gaps, but evidence for this in tropical forests is
mixed (Jansen et al., 2008). Given that our measurement intervals are relatively short (~one month), almost all of our mapped
canopy disturbances are likely to reflect single catastrophic events.
Our study is one of several that have documented size distributions of canopy disturbances (dynamic gaps) or of static
canopy gaps above some size that are approximately power functions, both on BCI (Solé and Manrubia, 1995; Lobo and Dalling,
2014) and in other tropical forests (Marvin and Asner, 2016; Asner et al., 2013; Kellner and Asner, 2009; Silva et al., 2019; Fisher
et al., 2008). Static canopy gaps are areas in which the forest canopy is below a threshold height, e.g., 10 m, at a given time.  A
power function distribution of disturbance event sizes (here canopy disturbances) and of the sizes of disturbed areas (canopy gaps)
can emerge from self-organization of dynamic systems such as forests (Solé and Manrubia, 1995). These same self-organized
dynamics lead to the development of equilibrium size distributions of trees, which are typically well-fit by Weibull distributions
in tropical forests (Muller-Landau et al., 2006b, a). The relative dearth of canopy disturbances smaller than 25 m$^2$ in our dataset,
compared to what would be expected under a power function, may be explained in part by lower detection frequencies. Our
methods are expected to capture all treefall and branchfalls above this threshold, but we may increasingly have missed smaller
events, especially below ~ 5 m$^2$. However, we consider it unlikely that this is a sufficient explanation for the shortfall in small
trees, and suggest that it is more likely explained largely by the low frequency of small trees and branches in the canopy of this
mature tropical forest, and thus a scarcity of small treefall and branchfall events.
Although rarely quantified, branchfall is an important ecological process, with major contributions to woody turnover and
necromass production. We found that branchfalls were almost as common as treefalls in number, although they contributed a
substantially smaller total area of disturbance. Similarly, a ground survey of 78 canopy turnover events in a Brazilian Amazon


forest found that 44 % were branchfalls, and that they accounted for 15 % of the total affected area (Leitold et al., 2018). In contrast,
a landscape level analysis of LiDAR data concluded that branchfalls were seven times more frequent than treefalls and accounted
for five times more area (Marvin and Asner, 2016). However, this study classified branchfalls and treefalls based purely on the
proportional decrease in canopy height (10-40 % decrease and 70-100 % decrease, respectively), a process liable to
misclassification, it entirely ignored disturbances involving intermediate decreases in canopy height (40-70 %), and did not
consider the possibility that any of these disturbances might be standing dead trees. Thus the contrast between our findings and
those of Marvin and Asner (2016) on the contributions of branchfalls may be due as much to methodological differences as to real
variation in canopy dynamics.

**5. Conclusions and future directions**

359       A mechanistic understanding of the controls on woody residence time in tropical forests is urgently needed to predict the
future of tropical forest carbon stocks and biodiversity under global change. Canopy trees account for a majority of the productivity
and carbon stocks in tropical forests, and their fates are disproportionately important for determining stand-level woody residence
time. Advances in drone hardware and photogrammetric software now make it relatively inexpensive and straightforward to
quantify forest canopy structure and dynamics at high spatial and temporal resolution through digital aerial photogrammetry and
repeat drone imagery acquisitions. Here we applied these methods to 50 ha of old-growth tropical forest for five years, and
analyzed the resulting products to quantify major drops in canopy height such as those created by branchfalls and treefalls, and
thus calculate the canopy disturbance rate. We found that canopy disturbance rates are highly temporally variable, and are well-
predicted by extreme rainfall events. Even higher temporal resolution canopy dynamics data together with higher frequency three-
dimensional wind data would enable an even stronger assessment of the link to storm conditions, and additional analyses of the
photogrammetry data could shed light on standing tree mortality. The expansion of these methods to additional and larger areas,
potentially in part through citizen science initiatives, has great potential to improve our understanding of tropical forest tree
mortality, and the future of tropical forests under changing climate regimes.

*Code    and    data    availability.*  Analysis    codes,    input    data    and    output    results    are    available    at
https://github.com/forestgeo/gap_dynamics_BCI50ha. All files will be published in a permanent form at Smithsonian Figshare
repository 10.25573/data.c.5389043 when the manuscript is published in final form.

*Author contributions.* HCM and RFA planned and designed the research. MG and JD collected drone data. RFA, SG, JD and MG
processed drone imagery. RFA performed the analysis with support from HCM, CHSC and RINJ. RFA and HCM wrote the
manuscript.

*Competing interests.* The authors declare that they have no conflicts of interest.






*Acknowledgments.* We gratefully acknowledge the financial support of the Next Generation Ecosystem Experiments-Tropics,
funded by the U.S. Department of Energy, Office of Science, Office of Biological and Environmental Research (RFA), the
Smithsonian Institution Competitive Grants Program for Science (HCM, JD), and the Smithsonian Tropical Research Institute
fellowship program (CHSC, RFA). We thank Milton Solano, Pablo Ramos, and Paulino Villareal for assistance in collecting and
processing the drone imagery, and Jeffrey Chambers, KC Cushman and Evan Gora for providing helpful comments on an earlier
version of this manuscript.

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
