# Peer review of "Strong temporal variation in treefall and branchfall rates in a tropical"

_Biogeosciences, 2021_

## Author Comment (AC1)

**Response to reviewer 1**

1) For the relationship between canopy disturbances and rainfall, I am worried if it makes sense to mention the 99.4th percentile in the Abstract because it may sound like cherry picking, like why '99.4' and not '99.3'? The correlation seems to change a lot in between 98-99 percentiles which may be a sign that the correlation may be spurious and not causal (Figure 5b). Moreover, if you look to Figure 5a and remove for example the most frequent rainfall event, the relationship would likely fall apart. How is the correlation for lower than 90 percentile? If the correlation would be causal I think it would be expected much weaker or no correlation at lower rainfall percentiles

*R: Thank you for pointing this out. We tested the analysis removing the highest canopy disturbance rate: the relationship remains significant and the highest Pearson decreases to 0.36 for the same 99.4$^{th}$ percentile. We also tested the correlations above the 80$^{th}$ percentile, and we found reduction in the Pearson values as the percentiles decreases, or even lose of significance (p-values around 0.1).*
*Following changes made in response to comments by reviewer 2, we have revised our analyses to use log-transformed data. The best predictor of temporal variation in canopy disturbance rates in the new analyses was the frequency of 1-hour rainfall events above the 97.7$^{th}$ percentile (r = 0.48). The residuals are better-distributed under log-transformation, and there is no longer any single point that exerts undue influence. (See responses to reviewer 2 for details of the revised analyses including the proposed revised figure.)*
*We propose to modify this sentence of the abstract to read: "The strongest correlate of temporal variation in canopy disturbance rates was the frequency of extreme rainfall events, here above 21.8 mm hour$^{-1}$ (r = 0.48)."*

2) I suggest authors consider adding a last paragraph of Discussion offering some advice for future studies, e.g. do you have recommendations for other researchers interested in replicating the experiments in other tropical forests, in regards to drone acquisition (camera, altitude, etc.), temporal frequency, etc. How would the replication of this study in other tropical forests help us understand the mechanisms better? This is a question to reflect and perhaps add something about these implications in this last paragraph. Some of these info is already scattered throughout the text but it could be important to have a concise paragraph on this.

*R: Thank you for the constructive criticism. We propose to revise and expand the section on Conclusions and future directions section to address these points (new text in blue below; the entire section is given for context):*

*"A mechanistic understanding of the controls on woody residence time in tropical forests is urgently needed to predict the future of tropical forest carbon stocks and biodiversity under global change. Canopy trees account for a majority of the productivity and carbon stocks in tropical forests, and their fates are disproportionately important for determining stand-level woody residence time. Advances in drone hardware and photogrammetric software now make it relatively inexpensive and straightforward to quantify forest canopy structure and dynamics at high spatial and temporal resolution through digital aerial photogrammetry and repeat drone imagery acquisitions. Here we applied these methods to 50 ha of old-growth tropical forest for five years, and analyzed the resulting products to quantify major drops in canopy height such as those created by branchfalls and treefalls, and thus calculate the canopy disturbance rate. We found that canopy disturbance rates are highly temporally variable, and are well-predicted by extreme rainfall events. Spatial resolutions of 3-7 cm in the orthomosaics, as used here, are now easily attained, and proved sufficient to capture canopy dynamics and visually classify disturbances as treefalls, branchfalls, or decomposition of standing dead trees.*

*Future research building on these approaches and expanding them to additional sites has much to contribute to our understanding of tropical forest dynamics. The relationship of standing dead tree mortality to temporal climate variation could be investigated from these same data by conducting additional analyses of the orthomosaics to quantify temporal changes in leafing status of standing dead trees, prior to these trees decomposing. A better understanding of the relationship of storm conditions to treefall and branchfall rates could be obtained by combining such drone-acquired data with mechanistic models of wind damage risk (Jackson et al. 2020), collecting higher frequency three-dimensional wind data, and/or measuring canopy dynamics at even higher temporal resolution. The use of drones with high accuracy GPS systems, either post-processed kinematic (PPK) or real-time kinematic (RTK) systems, would also be advantageous, and could enable elimination of the alignment step of the processing as well as automation of the identification of canopy disturbances based on elevation model differences alone. Finally, we recommend carrying out flights under cloudy conditions when possible, as these diffuse lighting conditions improve visibility deeper in the canopy and reduce complications associated with shadows. The expansion of these methods to additional and larger areas, potentially in part through citizen science initiatives, has great potential to improve our understanding of tropical forest tree mortality, and the future of tropical forests under changing climate regimes."*

3) In the Results/Discussion you say that you did not analyze the standing dead trees because you may miss those in your analysis. In the Abstract you suggest future studies of it. Perhaps in Discussion you could add some suggestion to better deal/analyze standing dead trees in future works.

*R: Good point. We have included a sentence on this in the proposed new concluding paragraph, which we present in response to the last point.*

4) L331-332, but did you find the effect of gap contagiousness? I was thinking about this when looking to the disturbances map, where lots of gaps were occuring nearby each other. Your data should allow you to test this hypothesis and likely is one of the best datasets around to do it.

R: We agree that our data provides a good opportunity to analyze gap contagiousness. We are analyzing this as part of another study we are conducting comparing patterns of canopy change between canopy gaps associated with treefalls vs. those associated with standing dead trees.

Technical corrections: L30, Strong -> robust?

R: Good point, we modified the wording.

L124, why put this in between parenthesis? it is useful infomation, should remove parenthesis

R: As suggested, we removed parenthesis. The text now reads: "We then pre-delineated major canopy disturbances by filtering for areas in which canopy height decreased more than 10 m in contiguous areas of at least 25 $m^2$, and that had an area-to-perimeter ratio greater than 0.6. We note that 25 $m^2$ is the minimum gap area used in previous studies of this site by Brokaw (1982) and Hubbell et al. (1999)."

L170, remove parenthesis – similar as before

R: As suggested, we removed parenthesis.

L172, what do you mean by "graphed"?

R: We propose to change the wording to "calculated".

L177, remove parenthesis – similar as before

*R: As suggested, we removed parenthesis.*

**L182, remove parenthesis – similar as before**

*R: As suggested, we removed parenthesis.*

**L235, Figure 5, It is a bit strange to show Pearson's correlation r besides a linear regression, it may misguide for R2**

*R: We agree. Our proposed revised figure does not include the regression line. Our analyses are based on Pearson correlations rather than linear regressions, and are now on log-transformed data, following changes made in response to suggestions from reviewer 2.*

**L352, this information about the criteria should be in methods**

*R: This information is about the methods Marvin and Asner (2016) used in their paper, not the methods of our study. We propose to reword for clarity (red highlights changed wording):*

*"In contrast, a landscape level analysis of LiDAR data concluded that branchfalls were seven times more frequent than treefalls and accounted for five times more area (Marvin and Asner, 2016). However, Marvin & Asner (2016) classified branchfalls and treefalls based purely on the proportional decrease in canopy height…"*

---

## Author Comment (AC2)

Response to reviewer 2

The authors present a unique analysis of canopy disturbances over the Barro Colorado Island 50-ha plot using a high-temporal density drone dataset. The high temporal resolution of this dataset allows the authors to relate the occurrence of canopy disturbance events to meteorological conditions with far greater precision than was previously possible with 5-year census intervals. The authors (surprisingly) conclude it is not horizontal wind speed, but high rainfall intensity events that cause canopy disturbances. Overall I think this is a very interesting analysis of a unique dataset, but I think it suffers from some analytical pitfalls that limit its utility for forest dynamics. I believe this will be a notable contribution if these issues can be addressed.

*R: We thank the reviewer for their positive comments. We note that we do not claim that rainfall causes canopy disturbances, but simply state that high rainfall is a better predictor than high windspeed in our analysis. As we noted in the abstract: "We hypothesize that extreme high rainfall is associated with both saturated soils, increasing risk of uprooting, and with gusts having high horizontal and vertical wind speeds that increase stresses on tree crowns." Anemometers may also have difficulty measuring windspeed accurately during heavy rain, and we propose to add a statement on this to the discussion.*

General comments:
There are some issues with the statistical analyses that I suggest be addressed (see line comments).
The size distribution of canopy disturbances is important. Table S3 seems like a really key piece of this study and should be in the main text. I suggest the authors include the equations of the distributions in the main text, and calculate some metric of uncertainty for each of the distribution parameters. It seems the lambda and k parameters of the Weibull distribution change quite a bit depending upon the minimum disturbance size. Although the Exponential distribution does not have the lowest AIC, the parameters don't shift as much.

*R: We are gratified by the reviewer's interest in the details of the size distribution analysis. We propose to revise the methods to include the equations for the distributions in the main text. The full text of this revised section is given later in this response. We also have now calculated 95% confidence intervals for the size distribution parameter values (by bootstrapping over the measurement intervals), added them to what was Table S3, and propose to move this table to the main text (replacing the current Table 1):*

*Table 1. Parameter values, Kolmogorov-Smirnov statistic, log-likelihood, and delta AIC values*

*for maximum likelihood fits of exponential, power and Weibull probability density functions to size distributions for canopy disturbances larger than 2 $m^2$, 5 $m^2$, 10 $m^2$ and 25 $m^2$. Delta AIC is the difference in AIC from the best model. The best-fit models for each dataset, and those within 2 delta AIC of the best model, are highlighted in bold.*

| Minimum size (m2) | Distribution | λ (95% CI) | α (95% CI) | K-S | Log likelihood | ΔAIC |
|---|---|---|---|---|---|---|
| **2** | **Exponential** | **0.018 (0.017 - 0.020)** | | **0.069** | **-4394.83** | **0.00** |
| 2 | Power | 1.312 (1.293 - 1.328) | | 0.340 | -5000.04 | 1210.42 |
| **2** | **Weibull** | **1.031 (0.938 - 1.199)** | **55.896 (50.148 - 64.230)** | **0.071** | **-4394.25** | **0.84** |
| 5 | Exponential | 0.019 (0.017 - 0.021) | | 0.067 | -4325.88 | 3.51 |
| 5 | Power | 1.480 (1.446 - 1.507) | | 0.271 | -4675.60 | 702.95 |
| **5** | **Weibull** | **0.922 (0.812 - 1.116)** | **48.915 (40.837 - 58.925)** | **0.057** | **-4323.13** | **0.00** |
| 10 | Exponential | 0.020 (0.018 - 0.022) | | 0.074 | -3995.34 | 16.82 |
| 10 | Power | 1.678 (1.644 - 1.713) | | 0.222 | -4174.91 | 375.95 |
| **10** | **Weibull** | **0.828 (0.735 - 0.997)** | **41.422 (33.920 - 51.297)** | **0.054** | **-3985.93** | **0.00** |
| 25 | Exponential | 0.020 (0.018 - 0.023) | | 0.100 | -2991.52 | 56.35 |
| 25 | Power | 2.165 (2.113 - 2.257) | | 0.081 | -2994.44 | 62.21 |
| **25** | **Weibull** | **0.532 (0.436 - 0.699)** | **12.271 (5.115 - 25.446)** | **0.022** | **-2962.34** | **0.00** |

*We also created two new figures (Fig. S7 and S8) comparing all the fitted distributions which we propose to add to the supplementary material (replacing Fig. S7 of the submitted manuscript). These figures illustrate how the different types of distributions compare in their fits for any one threshold (Fig. S7), and also how the fits for a given function differ depending on the minimum threshold (Fig. S8).*

[Figure]

*Figure S7. Observed size distributions of canopy disturbances, together with maximum likelihood fits under three alternative functional forms (exponential, power and Weibull functions). Each panel presents results for a particular minimum canopy disturbance area. Vertical dashed gray line indicates area thresholds.*

[Figure]

*Figure S8. Observed size distributions of canopy disturbances, together with maximum likelihood fits, compared for different minimum canopy disturbance areas. Each panel presents results for a particular type of fitted function: exponential (a), power (b) and Weibull (c).*

When calculating the hypothetical total canopy disturbance area from 1 million events, the Weibull and Exponential suggest near equivalent total disturbance area from the (fit 2m2) parameter set, but the Weibull only simulates 33% of the area simulated by the Exponential from the (≥25 m2) parameter set.

I see the authors used Python in the github repo (kudos for organizing the code), but in R it would be:

```
**Minimum size: 2 m^2**
**weibull and exponential agree**
sum(rweibull(1e6, scale = 55.860, shape = 1.03))/sum(rexp(1e6, rate=0.018))
**Minimum size: 25 m^2**
**The weibull fit simultes only 33% of the total from the exponential fit**
sum(rweibull(1e6, scale = 6.745, shape = 0.448))/sum(rexp(1e6, rate=0.02))
```

*R: We appreciate the reviewer's interest in calculating hypothetical canopy disturbance area, but note that the code included in the review draws from untruncated probability distributions, whereas our fits are for probability distributions truncated above at the maximum size that could have been observed, and truncated below at a minimum size to avoid small sizes at which we expect our methods to miss disturbances. We don't expect fitted distributions to necessarily behave similarly outside the truncated range that was fitted. Further we note that the reviewer's calculation of hypothetical total disturbance area are equivalent to calculating the mean disturbance size from the distribution, multiplied by the number of disturbances. We suggest that the mean size is more directly informative, and we show here in this response that the mean disturbance sizes of the truncated distributions are very similar between the fitted exponential and Weibull distributions and with the data in each case, although the fitted power function (Pareto) distribution has quite a different mean. We do this by modifying the reviewer's code as follows (although we note it can also be done analytically). We provide the output values from one realization in comments after each command.*

*library(EnvStats) # for the Pareto distribution, i.e., the power function distribution*
*nreps <- 1e6 # number of samples*
*maxgap <- 5e5 # maximum gap area possible in our study (50 ha)*

*# parameters for minimimum size 2 m2*
*mingap <- 2*
*weibshape <- 1.032*
*weibscale <- 55.93*

*exprate <- 0.01821*
*paretoshape <- 1.312 - 1*

*randgapweib <- rweibull(nreps, shape=weibshape,scale=weibscale)*
*randgapexp <- rexp(nreps, rate=exprate)*
*randgappow <- rpareto(nreps,location=minsize,shape=paretoshape)*

*# percentages of the distribution that are below the minimum size threshold*
*100\*pweibull(mingap,shape=weibshape,scale=weibscale) # 3.16%*
*100\*pexp(mingap,rate=exprate) # 3.57%*
*# none of the power function draws are below mingap because the minimum is one of the parameters of the Pareto*

*# percentages of the distribution that are above the maximum size that could have been observed*
*100\*pweibull(maxgap,shape=weibshape,scale=weibscale,lower.tail=F) # 0%*
*100\*pexp(maxgap,rate=exprate,lower.tail=F) # 0%*
*100\*(1-ppareto(maxgap,location=minsize,shape=paretoshape)) #2.07%*

*# mean gap area of the truncated distributions*
*mean(randgapweib[randgapweib>=mingap & randgapweib<=maxgap]) # 57.0*
*mean(randgapexp[randgapexp>=mingap & randgapweib<=maxgap]) # 56.9*
*mean(randgappow[randgappow>=mingap & randgappow<=maxgap]) # 4773*
*# for comparison, the mean size in the dataset is 56.9*

*# repeating for parameters for minimum size 25 m*
*mingap <- 25*
*weibshape <- 0.5326*
*weibscale <- 12.30*
*exprate <- 0.01982*
*paretoshape <- 2.165 - 1*

*randgapweib <- rweibull(nreps, shape=weibshape,scale=weibscale)*
*randgapexp <- rexp(nreps, rate=exprate)*
*randgappow <- rpareto(nreps,location=minsize,shape=paretoshape)*

*# percentages of the distribution that are below the minimum size threshold*
*100\*pweibull(mingap,shape=weibshape,scale=weibscale) # 76.8%*
*100\*pexp(mingap,rate=exprate) # 39.1%*
*# none of the power function draws are below mingap because the minimum is one of the parameters of the Pareto*

*# percentages of the distribution that are above the maximum size that could have been observed*
*100\*pweibull(maxgap,shape=weibshape,scale=weibscale,lower.tail=F) # 0%*
*100\*pexp(maxgap,rate=exprate,lower.tail=F) # 0%*
*100\*(1-ppareto(maxgap,location=minsize,shape=paretoshape)) # 0.000051 %*

*# mean gap area of the truncated distributions*
*mean(randgapweib[randgapweib>=mingap & randgapweib<=maxgap]) # 75.5*
*mean(randgapexp[randgapexp>=mingap & randgapweib<=maxgap]) # 75.4*
*mean(randgappow[randgappow>=mingap & randgappow<=maxgap]) # 139*
*# for comparison, the mean size in the dataset is 75.4*

If the end goal is to use these parametric distributions to estimate the total amount of canopy gap area being created, this discrepancy could have important implications for scaling. It would be nice to see a more thorough exploration of these distribution differences (and maybe check the Tweedie, Negative Binomial, LogNormal, Generalized Extreme Value dist.).

*R: Our aim in fitting the size distributions is not to estimate total amount of canopy gap area (or the mean gap size – we can obtain that directly from the data), but rather to evaluate the form of this size distribution. Most previous studies fit a single probability function to size distributions – the power function (Lobo and Dalling, 2013, 2014; Fisher et al., 2008, Asner et al., 2013; Kellner and Asner, 2009; Silva et al., 2019). We chose the power function, exponential distribution, and Weibull because these have been used to fit these or similar size distributions in the past (Muller-Landau et al., 2006a, Araujo et al., 2020, Higuchi et al., 2012). We recognize that there are many additional probability distributions that could be fit here, as is the case in general, but it is not typical in studies of this kind to explore all possible probability distributions. We further note that of the specific distributions suggested, the negative binomial is a distribution for discrete data, and thus is not appropriate in this case, and the form of the lognormal does not fit the data here. The Weibull provides a good fit, so we do not see a compelling argument to add additional distributions. Nonetheless, if the editor requests, we can add fits of particular additional distributions.*

I suggest along with the AIC, the log-likelihood also be presented.

*R: As suggested, we included the log-likelihood values in the proposed revised table (now Table 1).*

Apart from these, it would be useful to know which has the lowest mean

absolute error between the observations and (simulations) from the fit distributions, and which fit distribution produces the total simulated canopy disturbance area closest to the sum of the observations.

*R: We agree that additional measures to help readers understand the quality of the fit of the distributions would be useful. However, fitted and observed probability distributions are not usually compared in terms of mean absolute error. They are most often evaluated in terms of the Kolmogorov-Smirnov statistic for the maximum difference in cumulative probability between the observed and fitted distributions. We have added these statistics to our proposed revised Table 1, which was presented earlier in this response. As for the suggestion to include comparisons of the total disturbance area, as noted above, the expectation of the simulated total canopy disturbance area under a fitted distribution is equal simply to the mean times the number of simulated disturbances. We could add the observed and expected mean disturbance areas under each truncated dataset and fitted distribution to Table 1 if the editor thinks this would be worthwhile. We have not yet added it to the proposed revised table yet because we don't see this as a particularly good measure of fit, and are concerned the many different means (for different truncated distributions and fitted functions) could needlessly confuse readers*

Would the Weibull distribution still be the best fit distribution if the data were not binned (see: White, Enquist & Green 2008 Ecology)?

*R: Yes, we have now redone the fits without binning, and the ordering of the distributions is the same. The proposed revised Table 1 presented above shows results from fits without binning, which are qualitatively the same as before. We agree that fitting without binning is the better approach in this case and have revised methods and results accordingly (e.g., the results above are based on fits without binning). We originally binned the data because we adapted code from fits to diameter distributions, and tree diameter measurement data are essentially binned at the precision of the data (that is, e.g., a stem measured at 55 mm in reality has a diameter somewhere between 55.4 and 55.5 mm).*

It would be nice to see a histogram of the canopy disturbances on the raw untransformed scale, perhaps discretized by a few canopy depth classes. I suggest this could be added as a panel to one of the other figures. It would also be nice to see the canopy disturbance shapes in Fig 2 with a colorbar corresponding to the canopy depth. A 2D-density plot might be a way to present the distribution of the canopy gap size and depth.

*R: We appreciate this helpful suggestion. We have constructed a new graph along these lines, which we propose to add as a new panel c in Fig. 6, and which is shown below here. It is a stacked bar graph illustrating the distribution of canopy disturbances across area and height*

*drop classes. This graph clearly shows that canopy height drops increase with canopy disturbance size. We note that figure 6b of the submitted manuscript also presents information on the frequency of different combinations of gap area and depth, because we use transparency in plotting the points.*

[Figure]

I question the utility of reporting the canopy disturbance rate with respect to percentiles or thresholds, specific to the Barro Colorado Island met station. I urge the authors to reconsider this analysis with standard units (e.g. wind speed in m s-1, rainfall in mm hr-1). This would make the findings from this study more comparable with other studies, and potentially useful for parameterizing wind disturbance in ecosystem models.

*R: We agree that it is useful to translate the percentiles to the relevant thresholds in standard units. At the same time, we note that the analysis is most usefully done in terms of percentiles, because many precise windspeeds or rainfall rates are never observed, and all adjacent unobserved rates will produce exactly the same frequencies and thus the same correlation statistics. (For example, the 8th and 9th highest 1-hour rainfall rates observed are 49.0 and 45.7 mm. hour$^{-1}$, respectively, and thus all rainfall rates between these values will produce the same correlation statistics). We propose to add the following graph that shows how rainfall percentiles relate to rainfall rates in mm hour$^{-1}$ as a new panel in Figure 5 (proposed panel c).*

[Figure]

*We also propose to add the parallel graph for the windspeed analysis to SI Figure S6.*

[Figure]

On this topic, the max wind speeds in Figure S1 seem low - or is it the 7-day mean of the 15-min maximum? If so, it would be more useful to see the wind speeds unsmoothed because the effect of a strong storm gets washed out when averaged by week or month.

*R: We agree that it would be useful to show more information on the extremes of windspeed and rainfall and their variation. We propose to add two panels to Fig. S1 showing the daily maximum of 15-minute maximum wind speeds and 15-minute total rainfall. We have changed the units of rainfall to mm hour$^{-1}$, and we have modified the caption to better clarify what is graphed. The proposed revised graph and caption are as follows:*

[Figure]

*Figure S1. Temporal variation in rainfall and wind speed rates measured on Barro Colorado Island during the study period. Gray shading indicates the wet seasons (1 May to 31 December) of each year. (a) 1-day maxima of the 15-minute total rainfall. (b) 1-day maxima of the 10-second maximum wind speed. (c) 7-day and 30-day means of the 15-minute total rainfall. (d) 7-day and 30-day means of the 10-second maximum wind speed. We note that the windspeed measurements are taken every 10 seconds, with means, mininum and maxima of these measurements recorded every 15 minutes.*

Figure 6c is very interesting and odd. Could the plateau in frequency of the

smaller canopy disturbance area be related to a measurement bias? For example, perhaps all disturbances ≥25 m2 are visible from above the canopy, but perhaps smaller disturbances could be (partially) obscured by overtopping vegetation? Or could the canopy surface model not have sufficient resolution to identify smaller and shallower canopy disturbances on otherwise green canopies? Overall, I am not entirely convinced the plateau in Fig 6c is not caused by measurement bias.

*R: Indeed, we believe there is a high probability that the plateau below 25 $m^2$ is due in part to measurement bias, which is why we fitted distributions truncated below at 25 $m^2$. We addressed this in the discussion in lines 340-345 of the originally submitted manuscript:*
*"The relative dearth of canopy disturbances smaller than 25 $m^2$ in our dataset, compared to what would be expected under a power function, may be explained in part by detection bias. Our methods are expected to capture all treefall and branchfalls above this threshold, but we may increasingly have missed smaller events, especially below ~ 5 $m^2$. However, we consider it unlikely that this is a sufficient explanation for the shortfall in small trees, and suggest that it is more likely explained largely by the low frequency of small trees and branches in the canopy of this mature tropical forest, and thus a scarcity of small treefall and branchfall events."*

The following are suggestions that I hope the authors will consider addressing:

P1 L24: Confusing, power function and Weibull are very different.

*R: They are different over the entire distribution, but parts of Weibull distributions can be close to power functions. We propose to change the wording to: "The size distribution of canopy disturbances was best fit by a Weibull function, and was close to a power function for sizes above 25 $m^2$."*

P1 L26: Check units? (35.7 mm hour-1)

*R: We checked; the units are correct.*

P1 L29: "large spatial scales" ~ This seems relative. The spatial scale of this study is akin to the footprint of one MODIS surface reflectance pixel.
L30: confusing wording "linkages to drivers"

*R: We propose to reword: "These results demonstrate the utility of repeat drone-acquired data for quantifying forest canopy disturbance rates at fine temporal and spatial resolutions over*

*large areas, thereby enabling robust tests of how temporal variation in disturbance relates to climate drivers."*

L32: I suggest ending this abstract with a more conclusive statement about what was found, rather than a list of (potentially very difficult to accomplish) suggestions for other studies.

*R: We see one of the main contributions of our study being the demonstration of these methods, which have great potential to contribute even more to our understanding of canopy disturbances with some tweaks, which we see as entirely feasible to accomplish (indeed, we are working on pursuing all of these ourselves in ongoing work). We propose revising the wording to the following: "Further insights could be gained by combining such measurements with high frequency measurements of wind speeds and soil moisture to better capture proximate drivers, and incorporating additional image analyses to quantify standing dead trees in addition to treefalls." However, if the editor prefers, we can drop this sentence entirely.*

L35: The Pan 2013 reference is very old now, and was questionable to begin with. Surely there is a better reference at this point with the many radar/LiDAR RS studies?

*R: Thank you for pointing this out. We propose to change to referencing Xu et al. (2021).*

L38: Were either of these really theoretical? McDowell 2018 was more a review with a bit of speculation rather than a statement of theory, and Brienen 2015 presented a GAM of some sort for the Rainfor plots.

*R: Thanks for your suggestion. We removed the word "theory", and propose to change the statement to: "Tropical forest carbon stocks depend critically on tree mortality rates, and recent studies suggest tropical tree mortality rates may be increasing due to anthropogenic global change (Brienen et al., 2015; McDowell et al., 2018)."*

L40: I suggest placing the citation next to each disturbance (e.g. lightning strikes (Yanoviak et al., 2017), instead of lumping them together at the end.

*R: Thanks for your suggestion. We propose to change the statement to: "Tropical tree mortality can be caused by a diversity of drivers including storms (Fontes et al., 2018), droughts (McDowell et al., 2018; Silva et al., 2018), fires (Silva et al., 2018), lightning strikes (Yanoviak et al., 2017), and biotic agents (Fontes et al., 2018)".*

L43: I suggest referencing climate change rather than emissions scenarios,

which is the driver of climate change.

*R: We propose changing to: "An improved understanding of the processes of forest disturbance is critical to constrain estimates of current and future carbon cycling in tropical forests under climate change (Leitold et al., 2018)."*

L49-50: This seems surprising. What about following drought? At the very least, this statement is dependent upon the climate regime of the tropical forest in question.

*R: We understand that there are studies reporting higher mortality rates after drought periods in tropical forests (e.g. Zuleta et al.,2017 Drought-induced mortality patterns and rapid biomass recovery in a terra firme forest in the Colombian Amazon, Ecology). However, we aimed to compare with studies using fine temporal resolution (monthly and bi-monthly) measurement intervals in tropical forests and these three studies conducted in Panama and Central Amazon were the only ones we found in our search.*

L59: "easy" -> "easier"

*R: We changed in the text.*

L60: Suggest replace "stem density" with "stem basal area"

*R: The study we referenced reported that canopy trees constituted 40% of trees with DBH > 10 cm. It is a proportion of stem density, i.e., stems per area. Given the apparent potential for confusion, we propose to change the wording from "stem density" to "stems".*

L61: disproportionately useful to ...?

*R: We propose revised text: "Canopy trees constitute a high proportion of stems, aboveground carbon stocks and wood productivity (Araujo et al., 2020), and thus information on their mortality rates is disproportionately useful to understanding forest dynamics and carbon cycling."*

L62: I think it could be argued that windthrown but (temporarily) surviving trees will have reduced lifespans and their necromass is part of the "committed" emissions from necromass.

*R: That is very much the point we were trying to make. We propose to reword for clarity: "Treefalls do not necessarily result in tree mortality (trees may survive and resprout), but almost*

*all treefalls and branchfalls result in a large flux of carbon (wood) from biomass to necromass within a short time period after the event, which translates to reduced woody residence time."*

L65: "don't" -> "do not"

*R: We corrected the word in the text.*

L78: See paper "Death from above" by Deborah Clark. Branchfall might not be fatal to the tree losing the branch, but may be a large driver of understory mortality.

*R: Thanks for your comment. We propose to remove "non-fatal" from the sentence, which then reads: "Quantifying tree mortality and other damage such as branchfall contribute to a better understanding on change of forest structure, necromass estimates and nutrient cycling."*

L80: "5 years" -> "five years"

*R: Done.*

L83: "expect" or "hypothesize"?

*R: We propose to reword to "We expect that disturbance rates will be higher in the wet season than the dry season, we hypothesize disturbance rates will increase with the frequency of extreme rainfall and wind events, and we fit alternative models for predicting temporal variation in disturbance rates from rainfall or wind statistics."*

L94: decimal degrees might be better

*R: We changed coordinate format to decimal degrees.*

L96: Given that wind is an important part of this study, perhaps some statistics about wind gust speeds could be given (long term mean of max annual wind gust speeds, or some distribution?).

*R: As noted previously, we now present more information on maximum windspeeds in the proposed revised Figure S1. We calculated the average of the maximum daily wind speeds for dry and wet seasons (October 2014 to November 2019). The proposed revised text reads: "Mean of maximum 1-day wind speeds are 8.1 m s$^{-1}$ and 5.8 m s$^{-1}$ during dry and wet seasons, respectively."*

L106: So would a 1 second wind gust of 60 m/s have the same reading as a 14.9 minute sustained wind speed of 60 m/s? This might be an important point for the lack of a horizontal wind speed effect being found.

*R: No, we used maximum windspeeds not mean windspeeds. We propose revised text to more fully explain the wind speed measurements:"Wind speed measurements were made every 10 seconds, and the average, minimum and maximum values were recorded at the end of every 15-minute interval. We used the maximum wind speeds for our analyses."*

L126: "images for 1-ha square subplots" -> "images of 1-ha square subplots"

*R: We propose modifying the sentence to "Finally, we systematically examined 1-ha square subplots for each pair of successive dates and edited the pre-delineated polygons"*

L133: I suggest not using red to delineate the polygon on a green background because red/green is difficult for colorblind people to differentiate.

*R: Thank you for pointing this out. We changed the color of the canopy disturbance polygon to blue.*

L133: Minor issue: The Height bar goes from 162-186 m, but this is clearly not tree height. So maybe "Canopy Surface Elevation" would be more accurate?

*R: As suggested, we changed the legend to Canopy Surface Elevation.*

L149: I am unclear why the 237 day interval was excluded. Was this a data gap?

*R: Yes, this is a data gap - there were no image acquisitions during this time due to a drone crash and short-term lack of funds and personnel to recover from this setback. This time interval is almost three times larger than the next largest time interval in our dataset (91 days). We expect the data quality for this interval to be inferior to that for shorter intervals because the long time allows time for regrowth that hides evidence of disturbance. (We also switched drones and camera systems during this time.) We propose the following revised wording: "We excluded one excessively long interval (237 days – image acquisition gap) from all analyses of temporal variation".*

L160: Why linear regression as opposed to a glm or gam?

*R: We considered fitting more complex statistical models, but we were concerned to avoid overfitting, especially considering the limitations of the meteorological data and the fact that we have only 46 data points (time intervals), which are themselves not entirely independent (e.g., if one time interval had a strong storm that toppled many trees, then the a similarly strong storm in the next time interval might topple fewer trees because structurally unstable trees would already have come down, or it might topple more because some trees are now exposed to wind in ways they weren't before neighboring trees fell). We hope that the datasets we publish as part of the present study, combined with additional datasets, will provide material for our team and others to evaluate more complex models in the future.*

L172: with respect to the CDF plot, should this be referenced somewhere?

*R: Here we are explaining the data analyses; the relevant results figure is referenced in the results (but not in the methods), as is standard practice.*

L175-180: Are the size distributions being fit with all canopy disturbance drop heights? This would be a bit odd, as a canopy gap extending to the ground has different implications than say a shallow canopy gap that only extends 1 meter.

*R: Yes, the size distributions are fit to the areas of all canopy disturbances, regardless of height. This is why we refer to these as canopy disturbances rather than canopy gaps. We agree that canopy disturbances with different height drops have different implications for forest dynamics. The implications depend not only on the height drop, but also on the canopy height pre-disturbance. After all, a 15-m height drop might or might not extend to the ground, depending on the initial canopy height. As we note in the discussion, our canopy disturbance size distributions are not directly comparable with previously published canopy gap size distributions, which typically defined as continuous areas in which canopy height is below some value. A canopy disturbance event may or may not result in a canopy gap under a particular definition. A single canopy gap may represent one or more recent or older canopy disturbance events. The previous focus on canopy gaps was due in large part to their being easy to measure by people on the ground. In contrast, canopy disturbances are easy to measure with drones and other remote sensing, and are increasingly a focus of study (e.g., Marvin and Asner, 2016).*

L179: Unclear. Correlation with?

*R: We are correlating canopy disturbances height drop (m) and area ($m^2$). We thought this was clearly stated in the text: "We evaluated how average height drop was related to area across canopy disturbances, graphically and in terms of their Pearson correlation". If the editor*

*prefers, we can reword this, perhaps as follows: "We calculated the Pearson correlation between average height drop and area among canopy disturbances, and graphically evaluated how these were related."*

L180: Please include the functional forms of each distribution as equations in the main text. There are multiple forms of the power, and Weibull functions - so this will keep things clear.
L185: I suggest trying to explain this part in more detail. Most readers will not want to dig up the other paper to understand a core part of the methods for this manuscript.

*R: As suggested, we included the equations in the main text to improve clarity. We revised the text to more fully explain these methods. The proposed revised text reads:*

*"      We quantified the size distributions of canopy disturbances by fitting three alternative probability distributions: exponential, power (or Pareto), and Weibull (Eqs. 1-3, respectively).*

$$f_{exp}(x) = \frac{1}{N}\lambda e^{-\lambda x} \tag{1}$$

$$f_{pow}(x) = \frac{1}{N}x^{-\lambda} \tag{2}$$

$$f_{weib}(x) = \frac{1}{N}\frac{\lambda}{\alpha}\left(\frac{x}{\alpha}\right)^{\lambda-1}e^{-\left(\frac{x}{\alpha}\right)^{\lambda}} \tag{3}$$

*where $\lambda$ and $\alpha$ are fitted parameters, x is canopy disturbance area in $m^2$, e is the natural exponential basis, and N are normalization constants such that the truncated distribution integrates to 1. Recognizing that our methods are likely to miss smaller disturbances, we fit these distributions to truncated datasets, excluding disturbances below 2, 5, 10 or 25 $m^2$. Note that 25 $m^2$ is the minimum area for defining a canopy disturbance in our automated pre-delineation algorithm, and we are confident we captured all disturbances above this area. We also truncated the fitted distributions above at the maximum possible disturbance area we could have observed using our methods (50 ha, or 500,000 $m^2$). We fit each distribution to each dataset (different minimum disturbance area) using maximum likelihood. The maximum likelihood estimates of the parameters were those that maximized the likelihood function (Eq. (4)):*

$$L = \sum_i log[f(x)] \qquad (4)$$

*We selected the model that minimized Akaike's Information Criterion (AIC) (Burnham and Anderson, 1998). We also evaluated goodness of fit using the Kolmogorov-Smirnov statistic, the maximum difference in the cumulative probability distributions between the observed data and the fitted distribution (Carvalho, 2015)."*

L185: I suggest the log-likelihood also be presented (table 1).

*R: We included the log-likelihood in the revised Table 1, presented earlier.*

L188: suggest "last three years" -> "final three years of the time series"

*R: We now have canopy disturbances classified into treefalls, branchfalls and standing dead trees for all five years, with the exception of those that occurred during the long time interval. We modified the text to: "We classified each canopy disturbance as being a branchfall, treefall or standing dead tree decomposing, except for those disturbances occurring in the exceptionally long time interval. In 35 cases we could not distinguish the type of disturbance, and these cases were omitted from analyses that required disturbance classification."*

L190: I think the standing dead trees may be an issue for relating the tree falls to specific meteorological events. A standing dead tree may take years to fall, so it would be a misattribution to relate its death to a high wind speed event.

*R: We aimed to evaluate the contributions of rainfall and wind speed to canopy disturbance formation, not to tree mortality. Even if a tree is already standing dead, a storm can proximally cause the fall of this tree or its branches, creating new canopy disturbances. However, we can redo the analysis of canopy disturbance vs. rainfall and wind speed omitting standing dead trees if the editor requests.*

L187: Was there any field validation to determine if the branchfall and treefall classifications were correctly assigned?

*R: There was no on-the-ground field work to evaluate the classifications. The classification was visually assigned based on the temporal sequence of orthomosaics with 3-7 cm spatial*

*resolution, that give us highly detailed information on canopy dynamics. Examples are shown in the supplementary material in Figure S2. In most cases the images provided sufficient information to classify the cause of disturbance. However, as noted above, there were cases, especially in the first year when spatial resolution of the images was lower, when we were not able to classify the disturbance type from the images.*

L199: Is it possible to color code the branchfalls and treefalls (with a legend)?

*R: Yes, and we appreciate this suggestion. We created a new map colored by classes of treefall, branchfall and standing dead trees. We propose to replace Figure 2 in the main text with this figure, and move the current figure 2 (which distinguishes areas disturbed more than one time) to the Supplementary Material (Figure S9).*

[Figure]

L199: I suggest not using red to both outline the plot and indicate where two disturbances occurred.

*R: As suggested, we changed the color of plot boundary to black in relevant figures.*

L206: "parallel variation" is unclear.

*R: As suggested, we modified the sentence to: "There was strong temporal variation in canopy disturbance rates among the 46 time intervals analyzed, with similar temporal variation in the total area disturbed (Fig. 3) and in the number of disturbances (Fig. S4)."*

L215: I think the y-axis units are a bit misleading. It looks like the data gaps prevent analysis on a one month time step. For example, there is no way to know the monthly canopy disturbance rate around 2016 because the sampling interval is several months. Perhaps it is better to report the sum of disturbed area per sampling time block?

*R: We specifically chose the current graphing format to appropriately address the variation in the lengths of time intervals and avoid misleading readers. If we simply reported the total disturbed area in each time interval as the reviewer suggests, then longer time intervals would on average have higher total area, regardless of whether the disturbance rate (per time) were higher. By dividing the disturbance area by the time interval, we obtain the mean disturbance rate (per time) for each interval on the y, which is the quantity that will be of interest to most readers. We note that the horizontal axis is time, and that the bars for each interval have a width proportional to the size of the time interval. Thus the area of each bar is proportional to the total disturbance area. We have revised the caption to try to make this point more clear:*
  *"Rates are shown in units of percent of area per month, calculated as the sum of total area disturbed during the measurement interval, divided by the total area of the plot and by the length of the time interval in months (30-day intervals). Note that the total area of each rectangle is proportional to the total area of canopy disturbed during that measurement interval."*

L223: Why not present the early/late Dry season? Or better, put all in the same figure.

*R: We did not test for differences between the early and late dry seasons because there is no a priori reason to think these would differ, whereas prior publications and hypotheses do support differences between the early and wet season. Further, we note that sample sizes would provide little statistical power for such a test (just six observations in the early dry season and seven in the late dry season).*

[L223 continued]: I do not think the p-value adds much value here and it's calculation is not specified in the methods. Considering the skew in the data, the varied sampling intervals, and the intrinsic spatial dependency in the data, reporting simple p-values from (t-tests?) might not be statistically appropriate.

*R: The methods of the submitted manuscript clearly state "We tested for homogeneity of variances using the Levene test, and for differences between means using the two-tailed Student's t-test for the log-transformed canopy disturbance data." We've now also conducted the Shapiro-Wilk test for normality, and can confirm that the data do not violate assumptions of normality. We note that the statistic we are comparing is the disturbance rate in area per time period, which standardizes for differences in sampling interval length. As for intrinsic spatial*

*dependency – each point in this analysis is a single time interval, which encompasses many canopy disturbances. We propose to modify the methods section to mention the additional test for normality, and reword for clarity:*

*"We tested for differences in canopy disturbance rates between seasons using two-tailed Student's t-test on the log-transformed canopy disturbance rates for each measurement interval, after first confirming that these rates met assumptions for normality (Shapiro-Wilk test) and homogeneity of variance (Levene test)."*

*We also propose to add information on the source of the p-values to the figure caption:*

*"P-values are based on two-tailed Student's t tests for differences in log-transformed canopy disturbance rates between seasons."*

L235: Linear regression does not look like the right analysis for overdispersed data. It looks like the one large outlier exerts a lot of leverage to drive the r2 metric. I suggest the authors consider modeling this with a negative binomial or Tweedie generalized linear model.

*R: We agree that linear regression on untransformed data are not a good fit for these data. We have now conducted new analyses using Pearson correlations on log-transformed data. Residuals from linear regressions of log-transformed data are well-distributed, supporting the use of parametric Pearson correlations to summarize the relationship. The highlighted data point no longer exerts high leverage, and findings are qualitatively robust to its exclusion (even for the original analyses). Regarding the specific distributions suggested by the reviewer, we note that the negative binomial is a distribution for discrete data, whereas our response variable is continuous.*

*The relevant proposed methods text now reads: "We evaluated the relationship of temporal variation in canopy disturbance rates with temporal variation in climate extremes using linear regressions. We regressed the log-transformed canopy disturbance rates (area per time) against the log-transformed frequency of extreme rainfall and windspeed events (number per time)(i.e. $log(y)\sim log(x+1)$, for different definitions of extreme events."*

*The relevant proposed results text now reads: "The best predictor of temporal variation in canopy disturbance rates was the frequency of 1-hour rainfall events above the 97.7th percentile, which explained 23 % of the variation (Fig. 5a). This relationship was mainly driven by events occurred during wet seasons (Fig. 5a). This threshold outperformed all other tested rainfall thresholds (all percentiles from 90.0 to 99.9, by 0.1 % of the different frequency time scales – Fig. 5b). The 97.7th percentile corresponds to a rainfall rate of 21.8 mm hour$^{-1}$ (Fig. 5c). "*

L253: Why not use color in panel a?

*R: It is our view that gray and black are adequate to represent the cumulative distributions of canopy disturbances in terms of area and number. However we can change to using color if the editor so requests.*

L254: Is the correlation with height drop and canopy disturbance area, or the log of canopy disturbance area? I suggest the authors use a generalized additive model to overlay the trend on the points.

*R: Our proposed revised figure now includes a line from a generalized additive model (GAM) to illustrate the trend in the relationship, as suggested by the reviewer. Considering that this provides a good illustration of the relationship, we propose to omit mention of the Pearson correlation.*

[Figure]

L255: Should the exponential fit also be plotted?

*R: We aimed to compare the Weibull distribution (best fit) with the power distribution because the power function is widely used in the forest ecology literature to fit gap size distributions. As the exponential distribution had the worst fit for canopy disturbances > 25m2, we thought it not including it in the main text figure. We present it in SI instead. If the editor requests, we can add the exponential fit in the main text figure.*

L282: It might be worth noting that the horizontal wind speed was measured at ground level, and therefore might not really be representative of canopy surface wind conditions.

*R: Windspeed was measured at the top of the canopy, not at ground level. This is clearly stated in line 104 of the submitted manuscript that: "Wind speed was measured using an anemometer (RM Young Wind Monitor Model 05103) installed at the top of Lutz tower, at 48 m height above*

*ground and approximately 6 m above the top of the surrounding canopy."*

L295: High rainfall (mm), or high rainfall rate (mm hr-1)?

*Changed to rainfall rate.*

L327: The domino effect of falling trees causes spatial autocorrelation (effectively inflating
sample size), which ideally would be addressed in any of the regression analyses. In practice, this is difficult and would probably not change the conclusions of the manuscript.

*R: Yes, there are both spatial and temporal dependencies in the data that are not easily addressed. We hope that future efforts drawing on this dataset and others will succeed in accounting for these.*

L338: I am confused by what is meant by self-organization here. The wind storms are an exogenous force.

*R: We propose to add some additional words to explain this point: "A power function distribution of disturbance event sizes (here canopy disturbances) and of the sizes of disturbed areas (canopy gaps) can emerge from self-organization of dynamic systems such as forests in which individual tree growth and death depend on the sizes of neighbors (Sole and Manrubia 1995)." The cited paper, Sole and Manrubia 1995, explains this concept in detail, and shows how a simple cellular automata model can reproduce gap size distributions observed on BCI.*

L341: detection frequency -> measurement bias?

*R: Thank you for pointing this out. We changed the sentence to: "…may be explained in part by lower detection frequencies, i.e., measurement bias."*

L351-354: I suggest splitting this very long sentence in two.

*R: As suggested, we split the sentence. The text now reads: "However, this study classified branchfalls and treefalls based purely on the proportional decrease in canopy height (10-40 % decrease and 70-100 % decrease, respectively), a process liable to misclassification. It entirely ignored disturbances involving intermediate decreases in canopy height (40-70 %), and did not consider the possibility that any of these disturbances might be standing dead trees."*

L367: I am not sure about calling these 'rainfall events'. I suggest swapping

"extreme rainfall events" with "extreme storms". The trees are not falling down because of hard rain, they're falling because of the strong wind gusts accompanying these storms. The met station may be able to accurately measure rainfall intensity, but I think it's unlikely a 15-minute interval is going to be able capture the difference between sustained high wind speeds and very short gusts, so I think calling this "rainfall events" might be misattributing the cause to rain instead of wind.

*R: We agree. We changed the sentence to:*
*"We found that canopy disturbance rates are highly temporally variable, and are well-predicted by extreme rainstorms."*

L374: This is a unique and valuable dataset. Will both the raw and processed data will be published in the Figshare repository?

*R: Yes, we have uploaded all data to a Smithsonian Figshare repository, which will become public simultaneously with the publication of the final version of this manuscript.*

Fig S1: This is very surprising, the max wind speed never got above 7 m/s?

*R: The maximum wind speed did exceed 7 m/s. The previous graph showed the 1-day means of 15-minute maximum windspeeds. We now present 1-day maximum windspeeds in a new panel in Fig. S1; these peak at 12 m/s.*

Fig S2: Is the canopy gap disturbance counted as one polygon, or three separate polygons in panel F? Could these types of decisions have much influence on the distribution size fitting?

*R: All canopy disturbance polygons were considered individually. These three polygons are slow decaying branchfalls derived from the disintegration of a standing dead tree. We changed the caption to improve clarity: "...and disintegration of a standing dead tree – note that polygons were counted individually (e,f)." We note that standing dead trees represented only 8.2 and 9.2% of the canopy disturbance events and areas, respectively, and thus constitute a relatively small part of the dataset used for fitting size distributions.*

Fig S3: Why not present this as a color coded time series for each year of the study?

*R: Our aim with this figure is explain how we defined dry and wet seasons. We added more detailed rainfall information on Figure S1.*

Fig S7: I suggest adding the fit parameters for each distribution to the figure.

_R: We include these parameters in a main text table, which is referenced from the figure caption._